# Legal-Gated Attention Networks: Enforcing Action Legality as a Structural Inductive Bias in Deep Reinforcement Learning

## Abstract

In reinforcement learning (RL) for environments with state-dependent action constraints, conventional methods suffer from conflated representations, as signals from infeasible actions introduce noise and complicate the learning task. While post-hoc masking is a common workaround, it fails to prevent this contamination at a fundamental level, as illegal actions still influence the learned representations. To address this, we propose Legal-Gated Attention Networks (LGAN), an architecture that introduces a strong structural inductive bias by embedding action legality constraints directly into the attention mechanism. LGAN fundamentally alters self-attention by using a legality mask to gate the query formation process itself, permitting only legal actions to attend to the state. This architectural design guarantees that illegal actions are structurally eliminated: they produce no queries, receive no gradients, and cannot influence policy or value updates. By using raw state vectors as values, LGAN's attention weights directly reveal which state components contribute to each legal action's value. We demonstrate in board games that this structurally grounded approach provides an effective framework for learning transparent policies, positioning LGAN as a principled method for building robust and interpretable agents in action-constrained environments.

## 1 Introduction

Many real-world reinforcement learning (RL) environments impose state-dependent constraints on available actions—only a subset of the action space is executable in each state. Such legal action constraints arise in domains ranging from board games and program synthesis to robotics and combinatorial optimization. Ignoring these constraints leads to infeasible policies and unstable value estimation. In this work, we assume access to a per-state legality mask that indicates which actions are valid in each state. This assumption is realistic in many structured domains (e.g., board games, program synthesis), where legality can be directly derived from rules or simulators.

The core challenge stems from a structural mismatch: standard deep RL architectures are designed to approximate a mapping from a state to the entire action space, forcing them to process and represent all actions regardless of their feasibility. A common workaround is post-hoc masking, where the legality mask is applied externally during target computation or action selection. However, this approach is a superficial fix that fails to prevent a fundamental problem: representational contamination. Gradients from invalid actions still flow through and corrupt the shared network parameters. This contamination is particularly damaging for long-term credit assignment, where the high-variance noise from the vast space of invalid actions easily obscures faint and infrequent reward signals. Another example is for DQN. If illegal actions are not eliminated during the bootstrap step, this can lead to severe Q-explosion problems. The Bellman update's max operator may bootstrap from erroneously high Q-values of illegal actions, and even though these actions might be masked during final action selection, their inflated values have already contaminated the learned representations, causing systematic Q-value overestimation and learning instability. Furthermore, post-hoc masking creates a training–inference mismatch: the agent may learn to prefer an illegal action, but a late correction forces it to execute another, severing the link between the learned value and the executed action.

In this paper, we introduce Legal-Gated Attention Networks (LGAN), a Transformer-inspired architecture that structurally incorporates action feasibility into value and policy computation. Unlike standard self-attention, LGAN uses a novel gating mechanism where only legal actions form queries. This design not only enforces legality end-to-end without masking, but also yields an interpretable decision-making mechanism; since values are derived from raw state features via sparse attention, the model exposes attribution maps linking actions to state components. We evaluated LGAN across multiple domains with large and dynamic legal action sets, including Tic-Tac-Toe, Breakthrough, and Go, demonstrating that its structural inductive bias improves both learning performance and model transparency. Our contributions are:

- We propose LGAN, an attention-based architecture that structurally incorporates action feasibility into value and policy computation;

- We introduce LGAN's interpretable decision-making mechanism, based on sparse attention over raw state vectors;

- We evaluate LGAN across multiple domains with large and dynamic legal action sets, showing competitive performance and consistent attribution quality.

## 2 RELATED WORK

Early deep reinforcement learning (DRL) research, such as the seminal work in Atari games, primarily dealt with environments where all actions are legal Mnih et al. (2015). However, as DRL was applied to domains with dynamic action constraints, this issue became critical. The most common workaround is **post-hoc masking**, where a legality mask is applied to the network's final output (logits or Q-values), which ensures the policy is feasible. In value-based methods like DQN, masking illegal actions during the bootstrap step Mnih et al. (2015); Lanctot et al. (2019); Raffin et al. (2021) is a notable improvement over naively considering all actions Sutton & Barto (2018); Raffin et al. (2021); Liang et al. (2018), as it partially mitigates the Q-value explosion caused by representational contamination. Nevertheless, this remains a "superficial fix." Because the mask is applied at the end of the forward pass, gradients from illegal actions still flow backward and corrupt the shared network representations.

Other approaches tackle action constraints from different angles. Some methods attempt to learn legality from data, such as Action-Elimination DQN Zahavy et al. (2018) and Structured Mask Prediction Zhong et al. (2024), but illegal actions are still represented internally before elimination. Constrained MDP formulations Altman (1999); Tessler et al. (2018; 2019); Achiam et al. (2017); Yang et al. (2020) enforce feasibility via penalties or projections, but they primarily address soft constraints rather than the hard legality rules common in games or combinatorial tasks. Attention-based architectures for RL, such as GTrXL Parisotto et al. (2020a), focus on temporal stabilization and long-term dependencies, not on preventing contamination from illegal actions.

In summary, LGAN handles the legality problem by treating action legality as a structural inductive bias at the architectural level, structurally isolating contaminated representation, making it distinct from existing work.

## 3 METHOD

We consider a Markov Decision Process (MDP) defined by the tuple $(\mathcal{S}, \mathcal{A}, P, R, \gamma)$, extended to include state-dependent hard constraints on the action space. Each state $s$ has a corresponding real-valued feature vector $\boldsymbol{s} \in \mathbb{R}^d$ to represent the state information. At each state $s \in \mathcal{S}$, a subset of feasible actions $\mathcal{M}(s) \subseteq \mathcal{A}$ is available. This feasibility structure is represented by a binary vector $\boldsymbol{\mu}_s \in \{0, 1\}^{|\mathcal{A}|}$, where $\boldsymbol{\mu}_s[a] = 1$ if and only if $a \in \mathcal{M}(s)$. To construct a value function that inherently respects action constraints, LGAN integrates this feasibility mask directly into the attention mechanism: each legal action forms a **query**, a modified state vector serves as the **key**, and the raw state vector provides the **value**. This structure simultaneously enforces constraints and exposes the basis of each decision, offering both policy correctness and interpretability.

## 3.1 QUERY COMPUTATION

To ensure that only legal actions participate in decision-making, we gate the attention queries at the source. Each action $a \in \mathcal{A}$ is assigned a trainable embedding $\boldsymbol{e}_a \in \mathbb{R}^{1 \times d_{model}}$, which is optimized jointly with the rest of the network. The embedding matrix $E \in \mathbb{R}^{|\mathcal{A}| \times d_{model}}$ by stacks all action embeddings row-wise:

$$E = \begin{bmatrix} \boldsymbol{e}_1 \\ \boldsymbol{e}_2 \\ \vdots \\ \boldsymbol{e}_{|\mathcal{A}|} \end{bmatrix}$$

We use the legality mask $\boldsymbol{\mu}_s$ to zero out the embeddings of all illegal actions:

$$\textbf{query} = \text{diag}(\boldsymbol{\mu}_s)\, E$$

This gating ensures that only legal actions contribute to downstream computations. The learned action embeddings act as positional references in the action space, allowing the model to differentiate actions based on their semantic identity.

## 3.2 KEY AND VALUE COMPUTATION

We aim to preserve both spatial structure and interpretability in the way state features are represented. To incorporate positional information, we augment each scalar state feature $s_i$ with the standard sinusoidal positional encoding $\phi_i \in \mathbb{R}^{d_{\text{pos}}}$ as proposed by Vaswani et al. Vaswani et al. (2017). Rather than using a large, fixed base (e.g., 10,000) which can produce non-discriminative encodings in compact state spaces, we adopt an adaptive base, base $= 2 \times d$. This approach tailors the positional signal's frequency range directly to the state dimensionality $d$, ensuring a meaningful representation of spatial structure in board game environments. For our upcoming experiment on the game of Go, we employ a 3D positional encoding variant that concatenates separate embeddings for the layer, row, and column dimensions.

The augmented key vector is:

$$\tilde{k}_i = [s_i;\ \phi_i] \in \mathbb{R}^{1+d_{\text{pos}}}$$

By stacking the augmented vectors $\tilde{k}_i$ row-wise, we obtain the matrix $\tilde{K} \in \mathbb{R}^{d \times (1+d_{\text{pos}})}$, which is then linearly projected to form the full **key** matrix:

$$\textbf{key} = \tilde{K}\, W_k^{\top}, \quad W_k \in \mathbb{R}^{d_{model} \times (1+d_{\text{pos}})}$$

We use the raw state vector as the **value** directly, without applying additional nonlinear layers such as MLPs:

$$\textbf{value} = \boldsymbol{s} \in \mathbb{R}^d$$

While this limits the model's expressiveness, especially for more complex tasks, it ensures transparency by maintaining a clear relationship between state components and actions. Introducing MLPs would significantly reduce this interpretability, which is a trade-off we intentionally made.

## 3.3 ATTENTION COMPUTATION

We compute attention via a scaled dot product followed by a ReLU activation:

$$\textbf{Attention} = \text{ReLU}\left(\frac{\textbf{query}\,\textbf{key}^{\top}}{\sqrt{d_{model}}}\right)$$

where $d_{model}$ is the embedding dimension, this follows Vaswani et al. (2017) but replaces softmax with ReLU to enforce sparsity and zero-invariance. Given $h$ attention heads, we use a simple averaging as the aggregation strategy across the heads:

$$\bar{A} = \frac{1}{h}\sum_{i=1}^{h}\textbf{Attention}^i$$

The resulting $\bar{A}$ serves as a sparse, non-negative attribution over the state features.

### 3.4 INTERPRETABLE PREFERENCE SCORES

The preference score vector $z \in \mathbb{R}^{|\mathcal{A}|}$ is computed by applying the averaged attention map to the raw state vector:

$$z = \text{ReLU}(\bar{A}\,\textbf{value})$$

Each entry $z_a$ represents the aggregated influence of state features on action $a$, with $\bar{A}_{a,j}$ indicating how strongly action $a$ attends to the $j$-th state component. To ensure a straightforward attribution where attention weights directly correspond to the positive importance of state features, we make the assumption that all state features are non-negative (i.e., $\mathbf{s}_j \geq 0, \forall j$). This representation is common in many reinforcement learning environments, including the board games used in our experiments. This yields the decomposition:

$$z_a = \sum_j \bar{A}_{a,j}\,\mathbf{s}_j$$

which serves as a direct attribution: the impact of each state feature on the preference for action $a$ is made explicit and quantifiable. Because the attention is sparse and the value vector is untransformed, the interpretation remains transparent—no nonlinear mixing or hidden transformations intervene.

To prevent degenerate cases where legal actions are suppressed entirely (i.e., $z_a = 0$), we add a small constant:

$$z \leftarrow z + \epsilon\boldsymbol{\mu}_s, \quad \epsilon > 0$$

This ensures that all legal actions receive strictly positive scores, which is critical for downstream normalization in the actor and for avoiding $\log(0)$ in Q-value prediction.

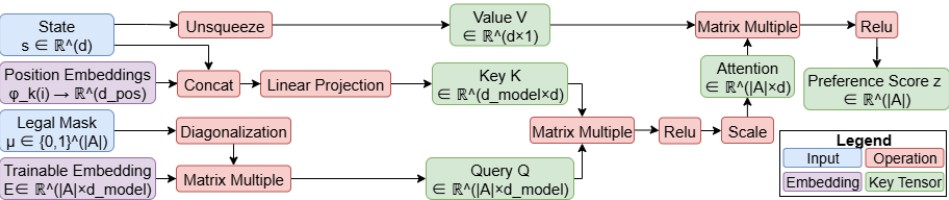

Figure 1: Partial computation flow (3.1-3.4) of LGAN. For clarity, the multi-head attention mechanism is not shown.

### 3.5 OUTPUT FOR REINFORCEMENT LEARNING

To apply the preference scores $z$—where $z_a \in (0, \infty)$ for legal actions and $z_a = 0$ for illegal ones—to standard reinforcement learning algorithms, we introduce a unified output mapping. This transforms $z$ into real-valued scores suitable for both value-based and policy-based methods. Specifically, we use a logarithmic function to define the final output, which serves as either a Q-value or a policy logit:

$$\text{Output}(s, a) = \log(z_a)$$

This operation projects the positive support of $z$ onto the entire real line $(-\infty, \infty)$, enabling unbounded regression. The logarithmic map is ideal as it is strictly increasing, thus preserving the relative preference ordering among actions: if $z_a > z_b$, then $Q_{\text{pred}}(s, a) > Q_{\text{pred}}(s, b)$. More importantly, it naturally handles illegal actions. Since an illegal action has $z_a = 0$, its Q-value becomes $\log(0) = -\infty$ (aligning with IEEE 754 standards), which automatically excludes it from any max-based operator without requiring additional masking logic.

**Q-value Computation**  The output is used directly as the predicted Q-value, $Q_{\text{pred}}(s, a)$. While the log transformation introduces a nonlinear scale distortion, we separate explanation from value prediction: interpretability is derived from $z$, while learning is performed over $\log(z)$.

**Actor Computation**  The output serves as the unnormalized logits for the actor's policy network. Although the logarithmic transformation alters the scale of the scores, its effect is canceled out by the subsequent softmax normalization. The resulting stochastic policy remains proportional to the original preference scores $z$, ensuring architectural consistency across different RL paradigms.

**Critic Computation**  For the critic in actor-critic algorithms, we compute the state value $V_{\text{critic}}$ directly from the **key** matrix, which encodes position-aware state features. Instead of using the raw state vector, we treat the transformed key as an enriched representation of the state. We apply a two-stage projection to compute the state value:

$$V_{\text{critic}} = w_2 \text{ReLU}(\mathbf{key}\, w_1)$$

where $w_1 \in \mathbb{R}^{d_{\text{model}} \times 1}$ compresses each state token into a scalar, and $w_2 \in \mathbb{R}^{1 \times d}$ aggregates these scalars across all positions. This design enables flexible value estimation while preserving structural awareness.

For a detailed analysis of the model's parameter complexity and total parameter count, please refer to the Appendix.

## 4 THEORETICAL ANALYSIS

### 4.1 ZERO-INVARIANCE AND LEGALITY PRESERVATION

We formalize LGAN's ability to enforce zero output for illegal actions purely through design, without relying on post-hoc masking.

**Theorem 1** (Zero-Invariance). *Let $s$ be a state and $a \notin \mathcal{M}(s)$ be an illegal action. Then, for any parameter setting, LGAN architecture guarantees:*

$$\boldsymbol{z}_a = 0$$

*Proof.* Recall that the query matrix is computed as $\text{Query} = \text{diag}(\boldsymbol{\mu}_s)\, E$, where $E$ contains the trainable action embeddings and $\boldsymbol{\mu}_s \in \{0,1\}^{|\mathcal{A}|}$ is the binary legality mask. For any $a \notin \mathcal{M}(s)$, we have $\boldsymbol{\mu}_a = 0$. This yields:

$$\text{Query}_{a,:} = \boldsymbol{\mu}_a \cdot \boldsymbol{e}_a = 0$$
$$\text{Attention}_{a,:} = \text{ReLU}(\text{Query}_{a,:} \text{Key}^\top) = \text{ReLU}(0) = 0$$
$$\boldsymbol{z}_a = \text{ReLU}(\text{Attention}_{a,:} \text{Value}) = \text{ReLU}(0) = 0$$

$\square$

Hence, the attention output $\boldsymbol{z}_a$ is deterministically zero for any illegal action, independent of model parameters, which ensures that outputs are supported only on legal actions, enabling interpretability and architectural correctness.

**Incompatibility with Softmax and BatchNorm**  Any architecture that applies softmax or batch normalization directly to $QK^\top$ violates zero-invariance. Softmax transforms zero vectors into strictly positive outputs. When applied across key dimensions for an illegal action ($Q_{a,:}K^\top = 0$), it produces a uniform distribution and assigns $\boldsymbol{z}_a > 0$. Applied across actions, softmax redistributes probability mass onto illegal actions, again breaking legality preservation. Batch normalization further disrupts zero-invariance by shifting and scaling logits based on global statistics, turning zeroed entries into nonzero values.

### 4.2 STRUCTURAL GRADIENT ISOLATION

The Zero-Invariance property not only ensures correctness during forward propagation but also confers a crucial benefit during the backward pass. This advantage, which we term Structural Gradient Isolation, guarantees that gradients are not propagated through illegal actions:

**Theorem 2** (Gradient Isolation). *Let $a \notin \mathcal{M}(s)$ be an illegal action. Then for any differentiable loss $\mathcal{L}$ and with respect to the embedding $e_a$, LGAN architecture satisfies:*

$$\frac{\partial \mathcal{L}}{\partial e_a} = \boldsymbol{0}$$

*Proof.* Since $z_a = 0$ is structurally guaranteed to be a constant 0, the chain rule yields:

$$\frac{\partial \mathcal{L}}{\partial \boldsymbol{z}_a} \frac{\partial \boldsymbol{z}_a}{\partial \boldsymbol{e}_a} = \frac{\partial \mathcal{L}}{\partial \boldsymbol{z}_a} \cdot 0 = \boldsymbol{0}$$

$\square$

This property ensures that the embedding parameters of illegal actions are excluded from updates. As a result, the model allocates its entire learning capacity to shaping representations of actionable behaviors. This structural gradient isolation safeguards the integrity of the Bellman update, prevents noise contamination at its source, and preserves the signal-to-noise ratio of long-term reward signals, thereby enabling more robust and faithful representation learning.

## 5 EXPERIMENTS

Our empirical evaluation is designed to answer several key research questions regarding the structural properties and performance of LGAN:

- **Correctness and Interpretability:** To verify that LGAN's architectural gating correctly confines the model's focus to the legal action set. And whether its attention mechanism produces verifiable, human-understandable insights into the decision-making process that align with known strategies.

- **Performance and Scalability:** To assess the performance and scalability of LGAN in relatively complex domains with large, dynamic action spaces. And compare its effectiveness against conventional models that have no structural inductive bias.

To address these questions, we selected three environments, each chosen for its distinct properties that facilitate one or more of our research goals:

- **Tic-Tac-Toe (TTT):** A simple, deterministic game with known optimal strategies. Its simplicity provides an ideal testbed for qualitatively verifying the correctness of the model's strategic choices and the interpretability of its attention-based attributions.

- **Breakthrough ($8 \times 8$):** A challenging combinatorial game with a highly dynamic legal action set (10-40 legal moves out of 768). It serves as our primary environment for evaluating LGAN's scalability and performance under significant action constraints, and its progressive nature makes it well-suited for standard RL.

- **Go (7x7):** A canonical benchmark with immense strategic depth and a vast state-action space. Its long games pose a significant credit assignment challenge, serving as a rigorous stress test for our architecture's scalability and effectiveness in a computationally demanding domain.

All models are trained via self-play under a sparse reward structure. Terminal rewards are $+1$, $0$, and $-1$ for wins, draws, and losses, respectively. Intermediate rewards are zero. A comprehensive list of common hyperparameters used across all experiments is provided in the appendix.

### 5.1 EXPERIMENT 1: CORRECTNESS AND INTERPRETABILITY ON TIC-TAC-TOE

The Tic-Tac-Toe (TTT) environment serves as a testbed to verify the correctness and interpretability of LGAN's decision-making process in a simple, deterministic game with known optimal strategies.

**Training and Evaluation Protocol** We trained separate LGAN agents as first and second players via self-play. Since both roles admit forced-draw strategies in Tic-Tac-Toe, the win rate against a random opponent (e.g., 95% as first player) is not informative. Thus, we focus on architectural interpretability rather than raw performance. We analyze a strategically interesting game state where the agent may identify two potential winning moves. Figure 2 displays the attention heatmap of a specified game state.

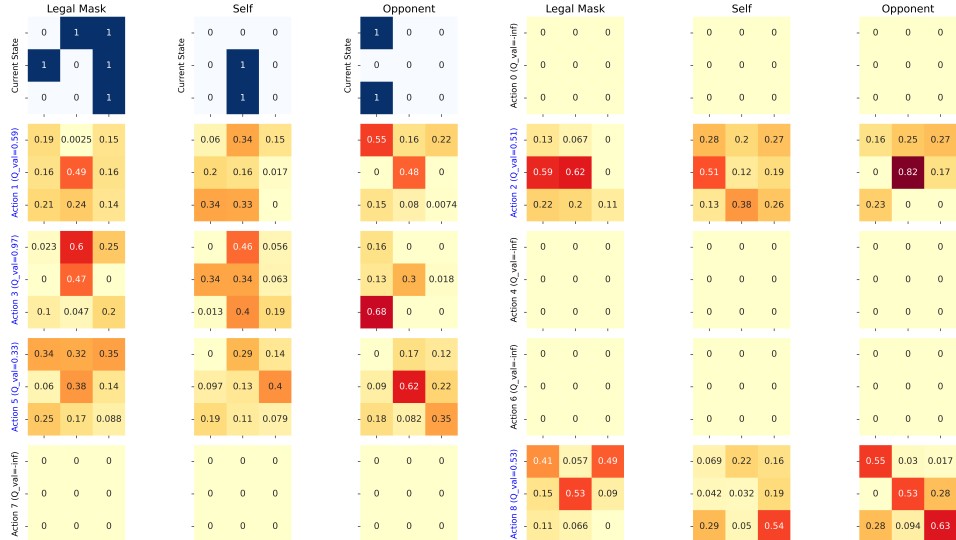

Figure 2: Attention heatmaps from a trained LGAN agent in Tic-Tac-Toe. The best action, Action 3 (Q-value=0.97), is primarily driven by strong attention to the opponent's piece at Position 6 (Opponent holds, contribution: 0.68), Position 1 (Blank, contribution: 0.60) and Position 7 (Self holds, contribution: 0.60). While the second best action, Action 1 (Q-value=0.59), is primarily driven by strong attention to Position 0 (Opponent holds, contribution: 0.55), Position 7 (Self holds, contribution: 0.33) and Position 4 (Self holds, contribution: 0.16). The heatmaps visualize the model's attention distribution over different state channels (e.g., self, opponent) for each action. High attention values (cells with deeper red color) indicate state components that contributed most significantly to the action's valuation. The attention corresponding to illegal moves is zero. Note that Action 0–8 correspond to board positions indexed from the top-left corner, proceeding row by row from left to right.

## 5.2 EXPERIMENT 2: PERFORMANCE ON BREAKTHROUGH

This experiment evaluates LGAN's generalization and performance in Breakthrough, a complex combinatorial game characterized by a highly dynamic legal action space. To specifically test the generalization capabilities of the attention mechanism itself, LGAN model in this experiment utilizes a simple 1D positional encoding. Consequently, to ensure a fair comparison focused on architectural advantages rather than complex feature engineering, we chose standard Multi-Layer Perceptrons (MLPs) as the foundation for our baseline models.

**Experiment Setup** We compare LGAN-DQN and LGAN-A2C against the following baselines: **MLP-DQN (Naive):** A standard DQN agent with an MLP Q-network. It allows illegal actions to influence bootstrapping. **MLP-DQN (Masked):** Identical to the naive version, but a legality mask is applied during bootstrapping to exclude illegal actions from the max operator. **MLP-A2C:** An Advantage Actor-Critic agent using separate MLP networks for the actor and critic. For baseline models, we apply a standard post-hoc legality mask by setting the logits of invalid actions to $-\infty$, which guarantees executability. In contrast, LGAN models structurally eliminate illegal actions.

Due to the strong player-order asymmetry in Breakthrough, we trained separate agents for the first and second player roles via self-play. Training proceeds for a number of self-play episodes, with performance periodically validated against a common reference opponent: an MCTS agent using 1000 simulations per move and uniform random rollouts (MCTS-1000). Training for each agent was terminated when its win rate against this MCTS opponent decreased. These evaluation matches against the MCTS agent were strictly used for performance tracking and were not added to the agents' training data.

Table 1: Win rates (%) against MCTS-1000 in Breakthrough.

| Agent | First Player | Second Player |
|---|---|---|
| LGAN-DQN | 97.2 | 95.5 |
| LGAN-A2C | 84.9 | 84.8 |
| MLP-DQN (Naive) | 0.0 | 0.0 |
| MLP-DQN (Masked) | 83.1 | 69.5 |
| MLP-A2C | 66.3 | 60.4 |

**Results and Analysis** The performance of all agents against the MCTS-1000 benchmark is presented in Table 1. The results clearly demonstrate the effectiveness of LGAN. In both algorithm categories, LGAN models significantly outperformed their MLP-based counterparts. Notably, the naive MLP-DQN agent completely failed to learn, exhibiting severe Q-value explosion throughout training. This highlights the critical importance of explicitly handling action constraints. Additionally, we conducted an ablation study on Breakthrough to validate our architectural choices; further details are provided in the Appendix.

## 5.3 Experiment 3: Performance on Go

The game of Go serves as a particularly challenging testbed for reinforcement learning due to its vast state-action space and the difficulty of long-term credit assignment caused by sparse terminal rewards. Acquiring fundamental Go concepts, such as forming "eyes" to secure living groups, requires an agent to connect actions taken early in the game to a sparse terminal reward, a task where standard deep learning models often fail without significant human priors or search algorithms.

**Experiment Setup** To isolate the effect of architectural inductive bias, we compare LGAN against strong, pure reinforcement learning baselines using a deep convolutional network with 6 residual blocks (ResNet), a powerful and standard architecture. Baselines include **RESNET-DQN** and **RESNET-A2C**. Our evaluation employs a cohesive, two-pronged strategy to provide a comprehensive assessment of performance. First, we measure absolute strategic competence by evaluating agents against a relatively weak MCTS benchmark(MCTS-200). MCTS is an ideal validator, as its search can ruthlessly exploit fundamental strategic weaknesses, providing a robust test of whether an agent has acquired core concepts. Second, we establish a definitive relative performance hierarchy by conducting a round-robin tournament among all trained agents.

**Results and Analysis** The results of the MCTS evaluation, shown in 2, reveal a stark difference in capability. The powerful ResNet-based agents completely failed to learn the game's core strategies, securing virtually no wins against the MCTS benchmark. This indicates a failure to solve the long-term credit assignment problem. In stark contrast, agents built on LGAN architecture were able to achieve meaningful victories, with LGAN-DQN demonstrating particularly strong performance as the second player (33.4 % win rate). This result suggests that LGAN has crossed a critical threshold of strategic understanding that the baselines could not reach.

Table 2: Win rates (%) against MCTS-200 in Go 7×7.

| Agent | First Player | Second Player |
|---|---|---|
| LGAN-DQN | 6.6 | 8.9 |
| LGAN-A2C | 14.6 | 16.2 |
| RESNET-DQN | 0.0 | 0.2 |
| RESNET-A2C | 0.0 | 0.1 |

The round-robin tournament, with results detailed in Table 3, confirms and strengthens these findings by establishing a clear performance hierarchy. LGAN architecture conferred a significant advantage, with both LGAN-DQN and LGAN-A2C consistently defeating the ResNet agents. LGAN-DQN

emerged as the tournament's top agent, demonstrating its dominance by holding even the strong LGAN-A2C to a mere 6.9% win rate when playing as the second player.

Table 3: Inter-agent win rates (%) in Go 7×7 (first vs. second).

| First \ Second | LGAN-DQN | LGAN-A2C | RESNET-DQN | RESNET-A2C |
| --- | --- | --- | --- | --- |
| LGAN-DQN | 100.0 | 62.1 | 100.0 | 80.9 |
| LGAN-A2C | 17.6 | 41.3 | 99.6 | 59.5 |
| RESNET-DQN | 0.0 | 0.4 | 100.0 | 0.0 |
| RESNET-A2C | 0.0 | 0.6 | 100.0 | 49.3 |

In a domain as complex as Go, the noise from gradients associated with the vast number of sub-optimal or illegal actions can completely overwhelm the faint signal from the terminal reward. A generic, high-capacity architecture like a ResNet is powerful but directionless, unable to distinguish this faint signal from the overwhelming noise. LGAN's Structural Gradient Isolation acts as a powerful filter, dramatically cleaning the learning signal by structurally zeroing out the influence of all illegal actions. This allows the agent to more effectively assign credit for the sparse terminal reward to the long sequence of legal actions that led to it. The Go experiment thus demonstrates that in the most challenging settings, a structural inductive bias like that in LGAN is not merely a helpful optimization; it can be a prerequisite for learning, enabling an agent to solve credit assignment problems where more powerful but less structured models fail.

## 6 LIMITATION AND FUTURE WORK

The direct, additive interpretability of LGAN, as presented, relies on the assumption of non-negative state features ($s_j \geq 0$). This ensures that a high attention weight corresponds to a positive contribution to an action's preference score. While this assumption holds for many RL environments, such as the board games used in our experiments where features represent piece presence, it limits the direct applicability of our attribution method to domains with signed or zero-centered features. A promising direction for future work is to extend this interpretability to general real-valued state spaces. One potential approach is a **dual-channel attention architecture**. The state vector $s$ could be decomposed into its positive and negative components, $s_{pos} = \text{ReLU}(s)$ and $s_{neg} = \text{ReLU}(-s)$. The model could then learn two separate attention maps, $A_{pos}$ and $A_{neg}$, to compute a preference score such as $z = \text{ReLU}(A_{pos}s_{pos} + A_{neg}s_{neg})$. This would allow the model to disentangle and explicitly represent the positive and negative evidence for each action, preserving transparency without constraining the input domain.

Another exciting avenue is the integration of LGAN with model-based planning: attention-derived preferences can guide action selection in an MCTS variant, potentially achieving top-tier performance. Finally, LGAN's binary legality mask may be relaxed to continuous values in $[0, 1]$, enabling soft gating where prior knowledge modulates attention strength across actions. This connects naturally to CMDP-style formulations, allowing the integration of graded feasibility signals into constrained but interpretable reinforcement learning.

## 7 CONCLUSION

We propose LGAN, an architecture that enforces action legality as a structural prior and enables transparent decision-making. By eliminating post-hoc masking and removing reliance on future legality signals, LGAN ensures correctness by design. Both theoretical analysis and empirical results support its robustness, interpretability, and extensibility—laying the foundation for constraint-aware and explainable reinforcement learning.

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

# A  APPENDIX

## A.1  PARAMETER ESTIMATION

The LGAN architecture enforces legality through structural design, requiring only a minimal number of learned components. Let $|\mathcal{A}|$ be the number of actions, $d$ the state dimension, and $d_{\text{model}}$ the embedding size. The total number of trainable parameters is given by:

- Action embeddings: $|\mathcal{A}| \times d_{\text{model}}$;
- Linear key projection: $(1 + d_{\text{pos}}) \times d_{\text{model}}$;

The overall parameter count for the core network is:

$$\text{Total parameters} = |\mathcal{A}| \cdot d_{\text{model}} + (1 + d_{\text{pos}}) \cdot d_{\text{model}} + h$$

In our experiments on an 8×8 Breakthrough task with $|\mathcal{A}| = 768$, $d = 192$, and $d_{\text{model}} = 512, d_{pos} = 24$, the total number of trainable parameters is **approximately 0.4 million**. Empirical cost 2.75GB GPU memory (batch size 128, including gradients).

This is significantly smaller than typical transformer-based RL architecturesParisotto et al. (2020b); Chen et al. (2021), yet sufficient for effective value approximation and policy extraction. The absence of softmax, batch normalization, and multi-layer value encoders further reduces computational overhead while preserving structural regularity.

If actor-critic methods are employed, the value function is obtained from the key matrix **key**. For the value head, the multi-head aggregation step is replaced by a parameter-free mean aggregation, thus removing the h aggregation weights. An additional linear projection is then applied to produce the value estimate, which introduces two parameter vectors:

$$w_1 \in \mathbb{R}^{d_{\text{model}}}, \quad w_2 \in \mathbb{R}^d$$

Consequently, the net parameter overhead is linear in the state and embedding dimensions, contributing a total of $d_{\text{model}} + d - h$ parameters.

## A.2  COMPUTATIONAL COMPLEXITY ANALYSIS

### A.2.1  TIME COMPLEXITY

Let $|A|$ denote the action space size, $d$ the state dimension, $d_{model}$ the embedding dimension, $h$ the number of attention heads, and $m(s) = |M(s)|$ the number of legal actions in state $s$. Forward Pass Complexity is given by:

- Query formation: **Query** $= \text{diag}(\mu_s) \cdot E$ requires $O(|A| \cdot d_{model})$
- Key computation: Positional encoding concatenation and projection via $W_k \in \mathbb{R}^{(1+d_{pos}) \times d_{model}}$ requires $O(d \cdot d_{model})$
- Attention scores: Computing $\mathbf{QK}^\top / \sqrt{d_{model}}$ for all heads requires $O(h \cdot |A| \cdot d \cdot d_{head})$ where $d_{head} = d_{model}/h$
- Value computation: $\mathbf{z} = \text{ReLU}(\bar{A} \cdot \mathbf{v})$ requires $O(|A| \cdot d)$

The overall parameter count for the core network is:

$$\text{Total complexity} = O(h \cdot |A| \cdot d \cdot d_{head} + |A| \cdot d_{model})$$

Since $d_{head} = d_{model}/h$, this simplifies to $T_{LGAN} = O(|A| \cdot d \cdot d_{model})$

## A.3  THE RATIONALE FOR THE INITIAL SELECTION AND EXCLUSION OF HYBRID AGENTS

The core scientific objective of this study is to isolate and evaluate the impact of encoding action legality as a structural inductive bias within a pure reinforcement learning (RL) framework. To ensure clarity of attribution, we deliberately restrict our comparisons to strong pure RL baselines

and exclude hybrid search-based agents, such as AlphaZero. Our chosen baselines, Multi-Layer Perceptrons (MLP) and Residual Networks (ResNets), operate entirely within the RL paradigm but lack the inductive bias of LGAN. This controlled setup allows us to attribute performance differences directly to architectural properties rather than external mechanisms.

By contrast, including AlphaZero-style hybrids would undermine the interpretability of our findings for two reasons:

- **Confounding Variable:** The performance of AlphaZero arises from the synergy of its Monte Carlo Tree Search (MCTS) and neural network components. Any performance gap between LGAN and AlphaZero would be inseparable from the overwhelming advantage conferred by MCTS, obscuring whether improvements derive from architectural design or search power.

- **Problem Misalignment:** LGAN is designed to address representational contamination within pure RL, where gradients from illegal actions destabilize learning. Hybrid agents largely sidestep this issue: MCTS restricts exploration to legal actions, effectively filtering the training signal before it reaches the network. Comparing LGAN to such systems would not meaningfully test its contribution, since the architectural problem it addresses has already been bypassed by external search.

In summary, our exclusion of hybrid baselines is a principled methodological choice. By focusing on pure RL architectures, we ensure a rigorous and unambiguous evaluation of LGAN's structural inductive bias in action-constrained environments. Moreover, hybrid search-based systems such as AlphaZero typically require orders of magnitude more computation and are not broadly applicable beyond a narrow set of domains, further underscoring the relevance of pure RL comparisons.

### A.4 HYPERPARAMETER DETAILS

The essential hyperparameters for our experiments are detailed in Table 4. To ensure full reproducibility of our results, we have made the complete configuration files publicly available in the project's code repository. All experiments are made on OpenSpiel.

Table 4: Hyperparameters used in the experiments.

| Parameter | Value |
|---|---|
| Optimizer | Adam |
| Discount Factor ($\gamma$) | 0.99 |
| Value loss function | MSE |
| **DQN-Specific** | |
| Replay Buffer Size | $1 \times 10^6$ |
| Initial Exploration Rate | 1.0 |
| Exploration Decay Duration | $1 \times 10^6$ steps |
| Final Exploration Rate | 0.1 |
| Train Interval | 10 steps |
| Target Network Update Interval | $1 \times 10^4$ steps |
| **Policy Gradient-Specific** | |
| GAE Lambda ($\lambda$) | 0.95 |
| Rollout Length | 128 |
| **TTT-Specific** | |
| Learning Rate | $3 \times 10^{-4}$ |
| Batch Size | 32 |
| LGAN Model Dimension | 512 |
| LGAN Number of Heads | 8 |
| **Breakthrough-Specific** | |
| Learning Rate | $1 \times 10^{-3}$ |
| Batch Size | 128 |
| LGAN Model Dimension | 512 |
| LGAN Number of Heads | 4 |
| LGAN Position Embedding dim | 24 |
| Baseline Model Dimension | 128 |
| A2C Value Loss Coefficient | 0.8 |
| **Go-Specific** | |
| Learning Rate | $3 \times 10^{-4}$ |
| Batch Size | 128 |
| LGAN Model Dimension | 512 |
| LGAN Number of Heads | 4 |
| LGAN Position Embedding dim | 24 |
| Baseline Model Dimension | 128 |
| Baseline Number of Blocks | 6 |
| A2C Value Loss Coefficient | 0.5 |

## A.5 ABLATION STUDIES

We conduct ablation studies on the Breakthrough 8×8 environment, reporting first and second player win rates separately. Unless otherwise specified, all models in this section adopt LGAN+DQN architecture with shared settings: $d_{\text{model}} = 512$, 4 attention heads, 1D positional encoding with base $2 \cdot d_{\text{model}}$, and Mean multi-head aggregation. Other training configurations remain consistent with those used in the main experiments.

### A.5.1 NUMBER OF HEADS

We evaluate the impact of varying the number of attention heads.

Table 5: Effect of number of heads in LGAN.

| Heads | First Player | Second Player |
|---|---|---|
| 2 | 81.1 | 84.2 |
| 4 | 92.3 | 85.1 |

As shown in Table 5, increasing the number of heads significantly improves performance. Multi-head attention enables the decomposition of different strategic patterns across heads, leading to more robust and generalized behavior under constrained capacity. Based on the performance, we choose 4 attention heads and move to the next state.

### A.5.2 POSITIONAL ENCODING STRATEGY

To assess the role of positional encoding (PE), we conducted an ablation study comparing learnable embeddings with fixed sinusoidal encodings.

Table 6: Impact of positional encoding strategy on the primary experimental platform.

| Encoding | First Player | Second Player |
|---|---|---|
| Sinusoidal-1D-16 | 92.3 | 85.1 |
| Sinusoidal-1D-24 | 97.2 | 95.5 |
| Sinusoidal-3D-24 | 80.5 | 75.4 |

We also experimented with a 3D positional encoding variant. For the game of Breakthrough, which is represented across three data layers, we allocated 4 PE dimensions to encode this layer information, with the remaining dimensions encoding the other spatial coordinates. However, this approach yielded poor performance.

Consequently, we adopted the 1D positional encoding strategy. A key consideration in our design is that the raw state feature is a boolean value with a vector length of one. Using a high-dimensional PE risks overwhelming this sparse input signal, which could drown out the original feature information and introduce training instability. Therefore, we deliberately selected smaller PE dimensions (e.g., 16 and 24) to ensure that the positional information complements, rather than dominates, the core state features, striking a balance between spatial awareness and model stability.

### A.5.3 USE OF LARGE LANGUAGE MODELS

Throughout the preparation of this manuscript, we utilized a large language model (LLM) as a writing assistant. The primary purpose of using the LLM was for proofreading and enhancing the clarity and readability of our text. The model was employed iteratively to help refine sentence structure, check for grammatical errors, and ensure that our ideas were expressed as clearly as possible. Its use was particularly significant in the **Introduction** and **Related Work** sections. From the **Methodology** section onward, the involvement of the LLM was minimal, as the content is predominantly based on our original technical contributions and experimental results. We, the authors, take full responsibility for all content presented in this paper, including any parts assisted by the LLM.

### A.5.4 REPRODUCIBILITY STATEMENT

We are committed to ensuring the reproducibility of our research. This work is self-contained and does not rely on external datasets. To facilitate replication, we provide a comprehensive table of all hyperparameters and architectural details in the preceding sections. In addition, we include the source code used in our experiments. However, the current implementation remains relatively unpolished, as it was primarily developed for rapid prototyping within a broader, ongoing research agenda.

Our larger investigation targets a central discrepancy in reinforcement learning (RL): why models with strong generalization capabilities, proven effective in other domains, often underperform compared to simpler architectures such as MLPs. We hypothesize that this underperformance stems from generalization across the boundary between valid and invalid actions in an environment's action space, a phenomenon we call *representational contamination*. To better understand this effect, the broader project is developing diagnostic tools to measure model sensitivity to representational contamination.

The model presented in this manuscript was originally conceived as a control study in this context, but its robustness to representational contamination led us to present it as a standalone contribution. While the released code may lack engineering refinements, it is sufficient to reproduce all reported experiments, and a more fully integrated version will be released as subsequent phases of our research are completed.

