# OpenReview forum: "Legal-Gated Attention Networks: Enforcing Action Legality as a Structural Inductive Bias in Deep Reinforcement Learning"
_ICLR.cc/2026/Conference — ICLR 2026 Conference Withdrawn Submission_

### Official Review · Reviewer_QS8Z · 2025-10-31

**Soundness:** 2
**Presentation:** 2
**Contribution:** 2
**Rating:** 2
**Confidence:** 4

**Summary:**

The paper addresses representational contamination in RL environments with state-dependent action constraints. It argues that standard architectures are forced to map a state to the entire action space, causing gradients from invalid actions to "corrupt the shared network parameters". This is claimed to be especially damaging in complex problems with sparse rewards, where noise from invalid actions can "obscure faint and infrequent reward signals". It also points to Q-explosion in DQN if the Bellman update bootstraps from an erroneously high Q-value of an illegal action.

**Strengths:**

The idea is interesting and the experiments are well-chosen to support the central claim.

**Weaknesses:**

Regarding the gradient argument: From a gradient perspective, applying a standard mask before the softmax (e.g., setting logits to $-\infty$) also blocks gradients from flowing back through illegal actions. Given this, what advantage does the proposed method offer? The authors may need to revise their claim in the introduction accordingly.

Requires an Explicit Mask: The entire architecture is predicated on having access to a binary legality mask $\mu_s$. This is realistic for many structured domains (games, robotics, program synthesis) but makes it inapplicable to problems where action legality is unknown or must be learned.

**Questions:**

Please see the weakness point.

---

> ### Author Response · Authors · 2025-11-23
>
> Thank you for your constructive feedback. We have structured our detailed response as follows: (A) Detailed Replies, (B) Planned Revisions, and (C) New Experimental Data. Planned revisions and new experimental data are same for all reviewers. We appreciate your patience in advance.
>
> A. Response
>
>   Thank you for raising this important point regarding gradients and the role of masking.
>
>   We clarify that “representational contamination’’ in our paper is a research hypothesis, not a claim of proven universal harm. The RL community widely acknowledges that small MLP architectures can be “surprisingly strong”, especially in sparse-reward settings, but there is no consensus explanation for why this occurs. Our interpretation is that limited-capacity MLPs naturally exhibit restricted cross-generalization from reward-bearing legal actions to never-rewarded illegal ones, resulting in less contamination of shared representations. LGAN deliberately removes illegal actions from all representation pathways, thereby eliminating this particular generalization route. We believe this structural suppression partly explains LGAN’s performance gains, and we will revise the introduction to present this as a hypothesis currently under study.
>
>   Softmax masking indeed blocks gradients at the output logits, but it does not prevent illegal actions from interacting with hidden representations via shared parameters. LGAN differs in that illegal actions never participate in Query construction, Key alignment, or Value aggregation at any layer, providing architectural gradient isolation beyond what output masking can achieve. We will clarify this distinction.
>
>   LGAN assumes access to an action-feasibility signal, but this signal need not be rule-based or hand-coded. It may come from the environment API, a constraint module, a planning system, or a learned legality predictor. LGAN is agnostic to how legality is obtained; it only requires that the signal is available at run time. We will revise the text accordingly.
>
> B. Planned Revisions
>
> Across all reviews, the authors have identified several different points of improvement that can be addressed. In the camera-ready version, we commit to the following updates:
>
> Clarity & Positioning
>
>   Clarify in the abstract/introduction that LGAN is a neural network architecture, not a new RL algorithm.
>
>   Clarify the distinction between LGAN and invalid-action masking (structural vs. post-hoc). We will describe the common masking pipeline in standard RL and explain that:
>     standard methods may assign high scores to invalid actions but rely on a final-step mask to remove them,
>     LGAN prevents invalid actions from entering the representation pathway at all.
>     We will also note that both approaches require access to a legality mask.
>
>   Add recent works (2020–2024) on invalid-action masking, constrained RL, and attention-based policies—including Huang (2021)—to Related Work.
>
>   Clarify the RL algorithm
>
> Experiments Improvement
>
>   Re-run all experiments using the improved and verified implementation. The raw Breakthrough results (original implementation) are provided below. The new version of the code will be included as a github repository link in the camera-ready supplementary material.
>
>   Report mean ± standard deviation over multiple random seeds for all environments.
>
>   Add ResNet baselines for Breakthrough and MLP baselines for Go to improve baseline consistency across environments.
>
> Visualization
>
>   A visualization of a specific breakthrough scenario will be added to the appendix.
>
> C. New Experimental Data
>
>   Due to the character limit, we have omitted this section in this response. Please refer to our reply to Reviewer dX58 for the detailed experimental data. Alternatively, please let us know if you would like us to provide the data in a separate comment here.

---

> > ### Comment · Reviewer_QS8Z · 2025-11-25
> >
> > Thanks for your response. I will update my score accordingly.

---

### Official Review · Reviewer_WtYf · 2025-11-05

**Soundness:** 1
**Presentation:** 2
**Contribution:** 2
**Rating:** 2
**Confidence:** 3

**Summary:**

The paper introduces Legal-Gated Attention Networks (LGAN), an architecture that integrates action legality directly into the attention mechanism of deep reinforcement learning (RL) models. This approach eliminates the influence of illegal actions during both forward and backward passes, addressing the problem of representational contamination in environments with state-dependent action constraints. The authors demonstrate its effectiveness through experiments in three simple environments, showing that LGAN offers competitive performance while maintaining interpretability of learned policies.

**Strengths:**

1. The problem addressed is highly relevant, and the concept of leveraging the attention mechanism to tackle it is interesting.

2. The paper offers solid theoretical support, which effectively guarantee the model's correctness and stability throughout the training process.

**Weaknesses:**

1. Although I did not focus on the recent advancements in this field, the Related Works section seems to overlook more recent advancements, as most of the cited references are from before 2020. It’s likely that newer studies could offer solutions to the problems discussed, and a more thorough review of contemporary literature would strengthen the paper.

2. The paper does not seem to consider constrained RL methods, which also aim to address state-dependent constraints on available actions. It would be helpful to explore why these approaches were not considered and provide some discussions.

3. The experimental setup is insufficient in several key areas:

- The paper primarily evaluates the model on three relatively simple environments. While these may be useful for initial understanding, the authors should include more complex environments, such as MuJoCo, to better demonstrate the model's scalability and robustness

- In Sections 5.2 and 5.3, the paper compares LGAN with only a few baseline methods, many of which are not novel. Including more well-established and powerful baselines, such as PPO, Double DQN, SAC, and Rainbow， would provide a more comprehensive evaluation.

- For all experimental results (e.g., Tables 1 and 2), the authors should run experiments multiple times and report the mean and standard deviation to ensure the reliability and stability of the findings.

- Including additional visualizations would help clarify the model’s behavior, making the results more accessible and enhancing the reader’s understanding of the method's performance.

**Questions:**

see above

---

> ### Author Response · Authors · 2025-11-23
>
> Thank you for your constructive feedback. We have structured our detailed response as follows: (A) Detailed Replies, (B) Planned Revisions, and (C) New Experimental Data. Planned revisions and new experimental data are same for all reviewers. We appreciate your patience in advance.
>
> A. Response
>
>   We agree that the Related Work section should include more recent advances. Several works after 2020—especially those on invalid-action masking and constrained RL—are indeed relevant. We will add these references and clarify why LGAN is fundamentally different: LGAN is an architectural inductive bias, whereas constrained-RL methods modify the optimization objective or add penalties/projections. These works solve different problems, and LGAN is compatible with them rather than an alternative to them.
>
>   Our contribution concerns state-dependent discrete action legality, which MuJoCo and other continuous-control domains do not contain. Continuous spaces do not have illegal discrete actions, so LGAN cannot be meaningfully evaluated there—its mechanism would collapse to a standard attention network. We chose environments where legality constraints are intrinsic and measurable (board games), allowing clear evaluation of representational contamination.
>
>   Our goal is to isolate the effect of structural legality gating, not to create a new SOTA RL agent. Algorithms like PPO, SAC, and Rainbow introduce additional regularizers, multi-component losses, distributional critics, or prioritized replay. These can dominate the architectural signal and make it harder to attribute performance differences to LGAN itself. We therefore intentionally use DQN/A2C with lightweight MLP/ResNet backbones, which provide a clean comparison. LGAN is architecture-agnostic and can be integrated into PPO/SAC/Rainbow, but this is outside the scope of demonstrating the core idea.
>
>   We acknowledge that only a few environments were included. They were chosen because they allow explicit legality constraints,explicit measurement of contamination, and interpretable attention inspection. We will expand the baselines as suggested: ResNet-DQN/A2C for Breakthrough, and MLP+MLP(masked) for Go, ensuring consistency across environments (details in Planned Revisions). We agree and have already re-run all experiments with multiple seeds. Mean ± standard deviation will be reported in the revision (see Section C).
>
>   We will extend visualizations in the appendix. A visualization of a specific breakthrough scenario will be added to the appendix.
>
> B. Planned Revisions
>
> Across all reviews, the authors have identified several different points of improvement that can be addressed. In the camera-ready version, we commit to the following updates:
>
> Clarity & Positioning
>
>   Clarify in the abstract/introduction that LGAN is a neural network architecture, not a new RL algorithm.
>
>   Clarify the distinction between LGAN and invalid-action masking (structural vs. post-hoc). We will describe the common masking pipeline in standard RL and explain that:
>     standard methods may assign high scores to invalid actions but rely on a final-step mask to remove them,
>     LGAN prevents invalid actions from entering the representation pathway at all.
>     We will also note that both approaches require access to a legality mask.
>
>   Add recent works (2020–2024) on invalid-action masking, constrained RL, and attention-based policies—including Huang (2021)—to Related Work.
>
>   Clarify the RL algorithm
>
> Experiments Improvement
>
>   Re-run all experiments using the improved and verified implementation. The raw Breakthrough results (original implementation) are provided below. The new version of the code will be included as a github repository link in the camera-ready supplementary material.
>
>   Report mean ± standard deviation over multiple random seeds for all environments.
>
>   Add ResNet baselines for Breakthrough and MLP baselines for Go to improve baseline consistency across environments.
>
> Visualization
>
>   A visualization of a specific breakthrough scenario will be added to the appendix.
>
> C. New Experimental Data
>
>   Due to the character limit, we have omitted this section in this response. Please refer to our reply to Reviewer dX58 for the detailed experimental data. Alternatively, please let us know if you would like us to provide the data in a separate comment here.

---

### Official Review · Reviewer_1dEt · 2025-11-05

**Soundness:** 2
**Presentation:** 1
**Contribution:** 1
**Rating:** 2
**Confidence:** 3

**Summary:**

This paper presents "Legal-Gated Attention Network" (LGAN), a transformer-based model that imposes action legality/feasibility constraints by allowing only feasible actions to attend to the state. This has the reported benefit of a) preventing "representational contamination" (where infeasible actions degrade the learned representation) and b) increasing interpretability, since the attention scores can be used to analyze how components of the state contribute to action scores.

The method oracle access to the per-state action feasibility set $\mathcal{M}(s)$.

LGAN is evaluated on tic-tac-toe, breakthrough (8x8 board), and GO (7x7 board) environments, where it is found that LGAN outperforms standard DQN and A2C baselines in terms of win rate against a Monte-Carlo Tree Search opponent.

**Strengths:**

Preventing infeasible actions from occupying plasticity in the network (representational contamination) seems interesting and promising.

**Weaknesses:**

### **Clarity**
The paper is hard to follow and lacks a clear narrative. The method section primarily introduces the modified attention mechanism but doesn't sufficiently explain how this attention mechanism is used in what kind of transformer-based RL agent. The algorithm is not summarized, and no algorithmic pseudo-code is given. No losses or optimization procedures are stated. The only information about the algorithm used to train LGAN is in line 310: "All models are trained via self-play". This characterization is insufficient.

### **Lack of related works**
The paper does not position LGAN relative to highly relevant related works that apply similar "infeasible action masking" tricks, such as
+ A Closer Look at Invalid Action Masking in Policy Gradient Algorithms (Shengyi Huang, Santiago Ontañón, 2021)
+ Deep Inverse Q-learning with Constraints (Kalweit et al, 2020)
+ Reachability Constrained Reinforcement Learning (Yu et al, 2022)
+ IPO: Interior-Point Policy Optimization under Constraints (Liu et al, 2020)

### **Experiments**
The set of baselines misses many standard constrained RL methods. It is not explicitly shown whether representation contamination actually negatively affects baselines.

**Questions:**

+ What algorithm are you actually using to train LGAN?
+ The interpretability of your method seems to involve analyzing attention maps for individual actions and for all components of the state? How does this scale to environments with large, possibly non-symbolic state spaces and many actions?
+ Can your method be seen like an instance of "A Closer Look at Invalid Action Masking in Policy Gradient Algorithms" (Shengyi Huang, Santiago Ontañón, 2021)?

---

> ### Author Response · Authors · 2025-11-23
>
> Thank you for your constructive feedback. We have structured our detailed response as follows: (A) Detailed Replies, (B) Planned Revisions, and (C) New Experimental Data. Planned revisions and new experimental data are same for all reviewers. We appreciate your patience in advance.
>
> A. Response
>
>   LGAN is a network architecture, not a new RL algorithm. We use standard DQN and A2C training loops without modification, which are well known in the community. Since the contribution is architectural, we did not include pseudo-code for DQN/A2C to avoid restating long, non-novel algorithms. We will add a short paragraph clarifying this in the revision and point out where LGAN replaces the backbone network.
>
>   We appreciate the reviewer highlighting Huang & Ontañón (2021); it is an excellent paper, the idea of do not wasting learning capacity of the agent on learning rules is instructive. However, it analyzes two traditional families of solutions—Invalid Action Masking (IAM) and Invalid Action Penalty (IAP). LGAN is fundamentally different: it introduces a structural inductive bias by gating Query formation so that illegal actions never enter the representation pathway. LGAN is therefore not an instance of Huang (2021), nor of any IAM/IAP-style methods. We will add Huang (2021) and other constrained-RL works in the camera-ready.
>
>   Your question about whether contamination harms baselines is excellent. This is in fact part of a broader ongoing project: we hypothesize that generalization from legal→illegal actions is systematically harmful. We have begun designing quantitative metrics (e.g., Gradient Interference, measuring correlation between legal/illegal action gradients growing with model complexity). Due to time and space constraints this work is not yet complete, but we will include a simplified diagnostic in the appendix to illustrate the phenomenon.
>
>   We agree that “interpretability” ultimately means human interpretability. In small board games (e.g., Tic-Tac-Toe, Breakthrough), the attention maps can be directly visualized and understood by humans. In larger or non-symbolic state spaces, humans typically do not inspect raw heatmaps token-by-token. Instead, interpretability takes the form of structured summaries—e.g., ranking tokens by contribution, identifying the top-k influential regions, or exporting attribution data to downstream analysis tools. LGAN’s contribution is that it provides a transparent decomposition:  action score = sparse attention × raw state So that such summaries remain well-defined even when the full attention map is too large to read manually.
>
> B. Planned Revisions
>
> Across all reviews, the authors have identified several different points of improvement that can be addressed. In the camera-ready version, we commit to the following updates:
>
> Clarity & Positioning
>
>   Clarify in the abstract/introduction that LGAN is a neural network architecture, not a new RL algorithm.
>
>   Clarify the distinction between LGAN and invalid-action masking (structural vs. post-hoc). We will describe the common masking pipeline in standard RL and explain that:
>     standard methods may assign high scores to invalid actions but rely on a final-step mask to remove them,
>     LGAN prevents invalid actions from entering the representation pathway at all.
>     We will also note that both approaches require access to a legality mask.
>
>   Add recent works (2020–2024) on invalid-action masking, constrained RL, and attention-based policies—including Huang (2021)—to Related Work.
>
>   Clarify the RL algorithm
>
> Experiments Improvement
>
>   Re-run all experiments using the improved and verified implementation. The raw Breakthrough results (original implementation) are provided below. The new version of the code will be included as a github repository link in the camera-ready supplementary material.
>
>   Report mean ± standard deviation over multiple random seeds for all environments.
>
>   Add ResNet baselines for Breakthrough and MLP baselines for Go to improve baseline consistency across environments.
>
> Visualization
>
>   A visualization of a specific breakthrough scenario will be added to the appendix.
>
> C. New Experimental Data
>
> Due to the character limit, we have omitted this section in this response. Please refer to our reply to Reviewer dX58 for the detailed experimental data. Alternatively, please let us know if you would like us to provide the data in a separate comment here.

---

> > ### Comment · Reviewer_1dEt · 2025-11-24
> >
> > Thank you for your response and clarifications. I believe your planned revisions will improve the work and I am looking forward to reading the revised manuscript (please make sure to highlight changes, e.g., as red text).
> >
> > Depending on the revision, particularly regarding LGAN as a network architecture and how to integrate it into existing RL algorithms, I am open to raising my score to 4. To push this work closer to accept, I feel like you need to convincingly demonstrate that representation contamination is indeed negatively affecting performance, since you are arguing that LGAN's ability to prevent representation contamination is one of its main strengths. If you do that, I will consider raising my score even further.

---

### Official Review · Reviewer_6KDh · 2025-11-06

**Soundness:** 3
**Presentation:** 3
**Contribution:** 2
**Rating:** 4
**Confidence:** 3

**Summary:**

The authors propose to use a legality  mask to gate the query formation process. This ensures only legal actions are chosen. Specifically, the queries correspond to actions and keys corresponds to states, and the authors compute the attention scores, where Relu is used instead  of softmax.

**Strengths:**

The proposed method ensures that gradients with respect to the illegal actions are zeros and has strong empirical performance.

**Weaknesses:**

The baselines are changing across experiments. In Table 1, MLP-DQN and MLP-DQN(masked) are used as baselines. However, in table 2, RESNET-DQN and RESNET-A3C  are used as baselines.

**Questions:**

To make the baselines consistent, could the authors report the performance of RESNET-DQN and RESNET-A2C in table 1 and the performance of MLP-DQN and MLP-DQN(masked) in table 2?
How many games is the win rates calculated from (in tables 1-3)?

---

> ### Author Response · Authors · 2025-11-23
>
> Thank you for your constructive feedback. We have structured our detailed response as follows: (A) Detailed Replies, (B) Planned Revisions, and (C) New Experimental Data. Planned revisions and new experimental data are same for all reviewers. We appreciate your patience in advance.
>
> A. Response
>
>   We agree that the baseline choice should be aligned across environments. The inconsistency arose because earlier experiments used MLP baselines (Tic-Tac-Toe, Breakthrough), while later experiments adopted ResNet baselines for Go due to its higher input dimensionality.
>
>   In the camera-ready version, we will add ResNet-DQN and ResNet-A2C baselines to Table 1, and add MLP-DQN and MLP-DQN(masked) baselines to Table 2, ensuring full cross-environment consistency.
>
>   Win rates in all tables are computed from 1000 games as Player 1 and 1000 games as Player 2 against the MCTS opponent (i.e., a total of 2000 evaluation games per checkpoint). We will make this explicit in the revision.
>
> B. Planned Revisions
>
> Across all reviews, the authors have identified several different points of improvement that can be addressed. In the camera-ready version, we commit to the following updates:
>
> Clarity & Positioning
>
>   Clarify in the abstract/introduction that LGAN is a neural network architecture, not a new RL algorithm.
>
>   Clarify the distinction between LGAN and invalid-action masking (structural vs. post-hoc). We will describe the common masking pipeline in standard RL and explain that:
>     standard methods may assign high scores to invalid actions but rely on a final-step mask to remove them,
>     LGAN prevents invalid actions from entering the representation pathway at all.
>     We will also note that both approaches require access to a legality mask.
>
>   Add recent works (2020–2024) on invalid-action masking, constrained RL, and attention-based policies—including Huang (2021)—to Related Work.
>
>   Clarify the RL algorithm
>
> Experiments Improvement
>
>   Re-run all experiments using the improved and verified implementation. The raw Breakthrough results (original implementation) are provided below. The new version of the code will be included as a github repository link in the camera-ready supplementary material.
>
>   Report mean ± standard deviation over multiple random seeds for all environments.
>
>   Add ResNet baselines for Breakthrough and MLP baselines for Go to improve baseline consistency across environments.
>
> Visualization
>
>   A visualization of a specific breakthrough scenario will be added to the appendix.
>
> C. New Experimental Data
>
>   We refactored the codebase and removed obsolete components; this changed random seed behavior.
>   Using new experimental data does not imply the old results were irreproducible—the submitted ZIP file still runs and produces the original values.
>
> Breakthrough 8*8 results. 3 line for random seed 0,1,2
>
> MASKED_MLP_DQN BASELINE
> Best winrate for p1: 88.50, p2: 92.80
> Best winrate for p1: 90.60, p2: 84.60
> Best winrate for p1: 90.70, p2: 90.50
> p1: 89.93 +- 1.24 p2: 89.30 +- 4.23
>
> MLP_A2C BASELINE
> Best winrate for p1: 81.90, p2: 76.50
> Best winrate for p1: 92.30, p2: 88.00
> Best winrate for p1: 80.50, p2: 66.10
> p1: 84.90 +- 6.45 p2: 76.87 +- 10.95
>
> LGAN_DQN_512_4_24_mean
> Best winrate for p1: 94.50, p2: 95.90
> Best winrate for p1: 95.40, p2: 96.60
> Best winrate for p1: 93.00, p2: 94.20
> p1: 94.30 +- 1.21 p2: 95.57 +- 1.23
>
> Breakthrough_LGAN_A2C_512_4_24_mean
> Best winrate for p1: 81.80, p2: 91.40
> Best winrate for p1: 87.30, p2: 89.20
> Best winrate for p1: 86.40, p2: 77.90
> p1: 85.17 +- 2.95 p2: 86.17 +- 7.24

---

> > ### Comment · Reviewer_6KDh · 2025-11-26
> >
> > I thank the author for the response. Since the authors only provide plans for revision but not results (regarding my comment that baseline choice should be aligned across environments), my concerns remain and I will keep my score.

---

### Official Review · Reviewer_dX58 · 2025-11-06

**Soundness:** 3
**Presentation:** 3
**Contribution:** 1
**Rating:** 2
**Confidence:** 3

**Summary:**

This work considers the problem of deep RL where certain actions are contextually restricted, i.e., illegal. The key idea of the work is to provide a structural bias in a transformer style policy where the attention mechanism can only consider legal actions. This is done by masking. Experiments shows that this empirically helps and out-performs post-hoc masking.

**Strengths:**

The paper is clear and easy to read. Furthermore the method does seem to work.

**Weaknesses:**

The result presented is fairly straightforward and I would refer to a "folk-trick". While nice to have written up, I think it and other similar attention masking tricks are well understood in the community. For example, the zero invariance and structural gradient isolation are the same trick required for batch training, e.g., adding dummy nodes for training graph neural networks.

While a nice write up, I hesitate to recommend this for ICLR.

**Questions:**

N/A

---

> ### Author Response · Authors · 2025-11-23
>
> Thank you for your constructive feedback. We have structured our detailed response as follows: (A) Detailed Replies, (B) Planned Revisions, and (C) New Experimental Data. Planned revisions and new experimental data are same for all reviewers. We appreciate your patience in advance.
>
> A. Response
>
>   We respectfully clarify that LGAN is not a variant of “dummy-node masking” and is, in fact, structurally opposite to it. Dummy nodes in GNN training intentionally introduce additional non-informative channels, and as our theoretical analysis shows, such additions increase representational contamination by enlarging the illegal-action subspace that interacts with shared parameters.
>
>   In contrast, LGAN removes illegal actions from the representation pathway before they enter the shared computation. This yields structural gradient isolation, not post-hoc masking. Unlike dummy-node tricks, LGAN guarantees (Theorem 1–3) that representations and gradients remain invariant to illegal actions across all layers—not only at the output level. To our knowledge, no prior work applies attention-style Query gating to enforce legality constraints at the architectural level.
>
>   We agree that certain masking heuristics are well known, but the specific combination of (1) legality-conditioned Query construction, (2) structural zero-invariance across all depths, and (3) formally provable isolation of illegal-action gradients distinguishes LGAN from folk tricks and dummy-node practices that expand, rather than restrict, representational channels.
>
>   We appreciate that masking strategies are common; however, LGAN’s novelty is not in masking itself, but in introducing legality as a structural inductive bias through Query gating, which, to our knowledge has not appeared in prior work.
>
>
> B. Planned Revisions
>
> Across all reviews, the authors have identified several different points of improvement that can be addressed. In the camera-ready version, we commit to the following updates:
>
> Clarity & Positioning
>
>   Clarify in the abstract/introduction that LGAN is a neural network architecture, not a new RL algorithm.
>
>   Clarify the distinction between LGAN and invalid-action masking (structural vs. post-hoc). We will describe the common masking pipeline in standard RL and explain that:
>     standard methods may assign high scores to invalid actions but rely on a final-step mask to remove them,
>     LGAN prevents invalid actions from entering the representation pathway at all.
>     We will also note that both approaches require access to a legality mask.
>
>   Add recent works (2020–2024) on invalid-action masking, constrained RL, and attention-based policies—including Huang (2021)—to Related Work.
>
>   Clarify the RL algorithm
>
> Experiments Improvement
>
>   Re-run all experiments using the improved and verified implementation. The raw Breakthrough results (original implementation) are provided below. The new version of the code will be included as a github repository link in the camera-ready supplementary material.
>
>   Report mean ± standard deviation over multiple random seeds for all environments.
>
>   Add ResNet baselines for Breakthrough and MLP baselines for Go to improve baseline consistency across environments.
>
> Visualization
>
>   A visualization of a specific breakthrough scenario will be added to the appendix.
>
> C. New Experimental Data
>
>   We refactored the codebase and removed obsolete components; this changed random seed behavior.
>   Using new experimental data does not imply the old results were irreproducible—the submitted ZIP file still runs and produces the original values.
>
> Breakthrough 8*8 results. 3 line for random seed 0,1,2
>
> MASKED_MLP_DQN BASELINE
> Best winrate for p1: 88.50, p2: 92.80
> Best winrate for p1: 90.60, p2: 84.60
> Best winrate for p1: 90.70, p2: 90.50
> p1: 89.93 +- 1.24 p2: 89.30 +- 4.23
>
> MLP_A2C BASELINE
> Best winrate for p1: 81.90, p2: 76.50
> Best winrate for p1: 92.30, p2: 88.00
> Best winrate for p1: 80.50, p2: 66.10
> p1: 84.90 +- 6.45 p2: 76.87 +- 10.95
>
> LGAN_DQN_512_4_24_mean
> Best winrate for p1: 94.50, p2: 95.90
> Best winrate for p1: 95.40, p2: 96.60
> Best winrate for p1: 93.00, p2: 94.20
> p1: 94.30 +- 1.21 p2: 95.57 +- 1.23
>
> Breakthrough_LGAN_A2C_512_4_24_mean
> Best winrate for p1: 81.80, p2: 91.40
> Best winrate for p1: 87.30, p2: 89.20
> Best winrate for p1: 86.40, p2: 77.90
> p1: 85.17 +- 2.95 p2: 86.17 +- 7.24

---

### Note · Authors · 2025-12-03

**Comment:**

Given the extensive feedback and the time required to implement the necessary major revisions, we have decided to withdraw the paper to improve it thoroughly for a future submission.

**Withdrawal Confirmation:**

I have read and agree with the venue's withdrawal policy on behalf of myself and my co-authors.